# TGF-Beta Induces Activin A Production in Dermal Fibroblasts Derived from Patients with Fibrodysplasia Ossificans Progressiva

**DOI:** 10.3390/ijms24032299

**Published:** 2023-01-24

**Authors:** Ruben D. de Ruiter, Lisanne E. Wisse, Ton Schoenmaker, Maqsood Yaqub, Gonzalo Sánchez-Duffhues, E. Marelise W. Eekhoff, Dimitra Micha

**Affiliations:** 1Department of Internal Medicine, Endocrinology Section, Amsterdam UMC, Amsterdam Movement Sciences, Vrije Universiteit Amsterdam, 1081 HV Amsterdam, The Netherlands; 2Department of Human Genetics, Amsterdam UMC, Amsterdam Movement Sciences, Vrije Universiteit Amsterdam, 1081 HV Amsterdam, The Netherlands; 3Department of Periodontology, Academic Centre for Dentistry Amsterdam (ACTA), University of Amsterdam and VU University, 1012 WX Amsterdam, The Netherlands; 4Department of Radiology and Nuclear Medicine Amsterdam UMC, Vrije Universiteit Amsterdam, 1081 HV Amsterdam, The Netherlands; 5Department of Cell and Chemical Biology, Leiden University Medical Centre, Universiteit Leiden, 2311 EZ Leiden, The Netherlands; 6Nanomaterials and Nanotechnology Research Center (CINN-CSIC), Health Research Institute of Asturias (ISPA), 33011 Oviedo, Spain

**Keywords:** TGF-beta, fibrodysplasia ossificans progressiva, fibroblasts, activin A, cytokines, BMP, inflammation, heterotopic ossification, ACVR1

## Abstract

Fibrodysplasia ossificans progressiva (FOP) is a catastrophic, ultra-rare disease of heterotopic ossification caused by genetic defects in the *ACVR1* gene. The mutant ACVR1 receptor, when triggered by an inflammatory process, leads to heterotopic ossification of the muscles and ligaments. Activin A has been discovered as the main osteogenic ligand of the FOP ACVR1 receptor. However, the source of Activin A itself and the trigger of its production in FOP individuals have remained elusive. We used primary dermal fibroblasts from five FOP patients to investigate Activin A production and how this is influenced by inflammatory cytokines in FOP. FOP fibroblasts showed elevated Activin A production compared to healthy controls, both in standard culture and osteogenic transdifferentiation conditions. We discovered TGFβ1 to be an FOP-specific stimulant of Activin A, shown by the upregulation of the *INHBA* gene and protein expression. Activin A and TGFβ1 were both induced by BMP4 in FOP and control fibroblasts. Treatment with TNFα and IL6 produced negligible levels of Activin A and TGFβ1 in both cell groups. We present for the first time TGFβ1 as a triggering factor of Activin A production in FOP. As TGFβ1 can promote the induction of the main driver of FOP, TGFβ1 could also be considered a possible therapeutic target in FOP treatment.

## 1. Introduction

Fibrodysplasia ossificans progressiva (FOP) is an ultra-rare genetic disease characterized by progressive ossification of skeletal muscles and associated soft tissues. Heterotopic ossification (HO) in FOP typically transpires through so-called flare-ups: episodes of soft tissue swelling accompanied by other classical features of inflammation such as pain, redness, and warmth [1]. The inflammatory response induces subsequent development of bone through endochondral ossification at the site of injury [2].

Approximately 97% of FOP cases are caused by a single nucleotide substitution mutation (c.617G > A, R206H) in the gene coding for the Activin receptor type 1 (ACVR1)/Activin receptor-like kinase 2 (ALK2), a bone morphogenetic protein (BMP) receptor [3,4]. This results in the replacement of arginine by histidine in the intracellular glycine-serine-rich receptor domain. However, even in patients with the same *ACVR1* mutation, the progression and severity of the disease can vary greatly [5]. It is well known that flare-ups can be triggered by environmental stimuli such as trauma, intramuscular vaccinations, and viral infections, though their presentation and evolution to HO can differ markedly per patient [5,6].

In this respect, the effect of inflammatory cytokines and growth factors is not to be overlooked in the pathology of FOP. In the earliest phases of a flare-up, there appears to be enhanced infiltration of macrophages and mast cells in the HO lesion [7]. In a conditional *ACVR1^R206H^* knock-in mouse model, post-injury lesions demonstrated increased immune cell infiltration, which also correlated with elevated and prolonged cytokine production of TNFα (tumor necrosis factor-alpha), IL1β (interleukin-1beta) and IL6 (interleukin-6) in the local lesion [7]. Depletion of the mast cells and macrophages resulted in significantly less injury-induced HO in this mouse model. In humans, blood samples taken from FOP patients without symptoms of a flare-up showed significantly elevated levels of pro-inflammatory interleukins [8,9]. Increased IL1β plasma levels were reported in an FOP patient (*n* = 1) with frequent highly active flare-ups, appearing to respond to treatment with anti-IL1 agents regarding reported flare-up incidence and IL1β levels [10]. On the other hand, systemic Activin A levels were not altered in individuals with FOP compared to healthy individuals, even when experiencing a flare-up [11]. Therefore, Activin A expression might be regulated locally, suggesting an important role for cytokines in the formation of FOP HO. However, a lot remains still to be determined about their source, spatiotemporal interactions, and microenvironment context in which they promote HO in the genetic background of FOP.

The transforming growth factor-β (TGFβ) superfamily of proteins has been established to be complex mediators of the immune system function [12]. Different BMP, Activin, and TGF-β proteins have been described to have both pro-and anti-inflammatory activity in different disease and cellular contexts [13,14]. TGFβ superfamily proteins also play an integral role in the mechanism of FOP. The ACVR1 receptor is triggered by BMPs, leading to an osteogenic response through canonical SMAD 1/5/9 signaling [15]. Different modes of activation have been reported, with the R206H ACVR1 receptor appearing to be constitutively active and hyper-reactive to BMP signaling [16]. Perhaps more significant towards the pathophysiology of FOP, the mutated ACVR1 receptor also responds to Activin A to similarly trigger SMAD 1/5/9 signaling, driving subsequent HO [17]. Inhibition with an Activin A antibody showed almost complete inhibition of HO in an FOP mouse model, inspiring further development of anti-Activin A neutralizing antibodies as a potential therapy for FOP [18].

Limited evidence exists on the role of TGFβ regarding HO in FOP. Our group has already demonstrated that the chemical inhibition of TGFβ-like signaling in a model of osteogenic transdifferentiation was able to abolish cell differentiation of FOP fibroblasts in the absence of recombinant ligands [19,20]. Resembling the reported variability in FOP progression, FOP cell lines showed a varying capacity for osteogenic transdifferentiation [19]. We hypothesized that this is perhaps due to variable production of cytokines such as Activin A promoting osteogenic transdifferentiation in the different patient cell lines.

Considering the prominent contribution of inflammatory tissue responses in FOP progression, it stands to reason that the composition of cytokines and other growth factors of the inflammatory niche can be decisive in determining the cell fate of FOP osteogenic progenitors. In this study, we aimed to gain further insight into the molecular mechanisms by which inflammatory factors modulate Activin A production. To this end, we investigated the role of TGFβ1, TNFα, and IL-6 on Activin A and TGFβ1 production and expression of downstream TGFβ superfamily target genes in dermal fibroblasts derived from patients with FOP and healthy controls.

## 2. Results

### 2.1. SMAD Signalling in Fibroblasts of FOP Patients

SMAD1/5/9 phosphorylation was investigated in cultured primary fibroblasts derived from FOP patients and controls. The classic *ACVR1* c.617G>A R206H mutation was confirmed by Sanger sequencing in the patient cell cultures (Appendix A). After overnight serum starvation, control and FOP patient fibroblasts both exhibited SMAD1/5/9 phosphorylation after BMP4 stimulation, and no differences between the FOP and control groups were observed (Figure 1). Activin A stimulation led to SMAD1/5/9 phosphorylation in FOP but not in control fibroblasts, confirming the reported responsiveness of the mutated ACVR1 receptor to Activin A in this cell model [19,20].

### 2.2. Activin A Production by FOP Fibroblasts

In a previous study, we reported the increased potential for osteogenic transdifferentiation in the FOP fibroblasts. Fibroblasts were differentiated towards cells of an osteogenic lineage (osteoblast-like cells) and were validated regarding gene expression and mineralization assays [19]. Interestingly, this potential varied between the patient cell lines; a trend of higher osteogenic transdifferentiation was observed in three of the five FOP fibroblast lines (P2, P4, and P5), as shown by the increased expression of the *RUNX2* and *ALP* osteogenic markers [19]. Considering the role of Activin A as a known driver of osteogenesis in FOP, its production was measured by ELISA in FOP fibroblasts supernatant in standard culture conditions. We found increased Activin A production in FOP fibroblasts compared to control cells (Figure 2A) (unpaired t-test, *p* < 0.05), though the production gradually declined over the first three days (Figure 2B). This difference was also present during the osteogenic transdifferentiation of the fibroblasts.

### 2.3. Cytokine Regulation of Activin A Production by FOP Fibroblasts

The higher production of Activin A by FOP fibroblasts prompted us to investigate its regulation. Given the contribution of the inflammatory micro-environment in FOP lesions, Activin-A was quantified by ELISA after 24 and 48 h of stimulation with several disease-relevant recombinant cytokines such as TGFβ1, BMP4, TNFα, and IL6 (Figure 3A). Stimulation with TGFβ1 resulted in significantly higher production of Activin A in FOP fibroblasts, both at 24 and 48 h of stimulation. Interestingly, the Activin A production was the highest in patients P2, P4, and P5. Stimulation with BMP4 also produced the same effect, although in this case, significant upregulation of Activin A production was also shown at 24 h. Again, P2, P4, and P5 were the most responsive patients to BMP4 stimulation, as shown by the higher Activin A production. Stimulation with TNFα did not lead to significant differences in Activin A production, and most cell lines showed levels close to the sensitivity limit of the ELISA (12 pg/mL). Similarly, low levels of Activin A were produced after stimulation with IL6 in all conditions (Appendix A).

Considering the clear stimulatory effect of TGFβ1 on Activin A production, we investigated to which extent TGFβ1 secretion could be, in turn, affected by cytokines. To this end, FOP and control cells were stimulated with Activin A, BMP4, and TNFα for 24 h and 48 h (Figure 3B). Stimulation with Activin A did not reciprocate the effect on TGFβ1, as shown by the lack of statistically significant differences in TGFβ1 production; most cell lines produced TGFβ1 levels close to the sensitivity limit of the ELISA (80 pg/mL). On the contrary, BMP4 treatment led to significantly increased production of TGFβ1 in control cells at 24 h and in both FOP and control cells at 48 h. TNFα stimulation also resulted in low response in TGFβ1 production in all conditions, and vice versa, no detectable production of TGFβ1 was observed after TNFα stimulation (Appendix A). As no differences were observed after TNFα or IL-6 stimulation, only the effects of TGFβ and Activin A were validated on a gene expression level.

### 2.4. Expression of BMP and TGFβ Target Genes

Stimulation of the FOP ACVR1 receptor leads to the expression of BMP target genes downstream of pSMAD1/5/9 signaling [21]. Thus, we investigated if that is the case after stimulation with the cytokines Activin A, BMP4, and TGFβ1 for 6 h (Figure 4A). As expected, stimulation with Activin A produced a significant increase in the relative gene expression of *ID1* and *ID2* (two-way ANOVA, *p* < 0.05) in FOP compared to control fibroblasts. BMP4 stimulation produced the same effect in both FOP and control cells, in agreement with the lack of differences in pSMAD1/5/9 expression (Figure 1A). Interestingly, stimulation with TGFβ1 also led to a significant upregulation in *ID1* expression in FOP cells (two-way ANOVA, *p* < 0.05), although this was not observed for *ID2*. No significant differences were found for BMP target genes *DLX1* and *MSX2* (Appendix A). Relative gene expression of the TGFβ target genes *CTGF*, *COL1A1*, and *THBS1* was also measured in the same conditions (Appendix A). Treatment with Activin A did not produce significant changes in *CTGF* expression, whereas treatment with BMP4 did lead to a significant *CTGF* increase (two-way ANOVA, *p* < 0.05), in agreement with the increased TGFβ production after BMP4 stimulation (Figure 3B). As expected, TGFβ stimulation led to significantly higher expression of *CTGF* both in FOP and control cells (two-way ANOVA, *p* < 0.05); no specific gene expression pattern was observed for *COL1A1* and *THBS1*.

### 2.5. Gene Expression of Cytokines in Fibroblasts

Based on these findings, it is clearly shown that TGFβ1 triggers Activin A production in FOP, and not control, fibroblasts at both 24 and 48 h, whereas after stimulation with BMP4, Activin A production was increased in both FOP and control cells at 24 h and remained high in FOP cells at 48 h. In order to address the possibility of feedback mechanisms, the relative expression of *INHBA, TGFBR1, IL6*, and *IL1b* was investigated (Figure 5). In line with the above, stimulation with TGFβ1 and BMP4 led to significantly increased *INHBA* expression in FOP cells for TGFβ1 and both cell types for BMP4 (two-way ANOVA, *p* < 0.05).

## 3. Discussion

In FOP, inflammation begets ossification when mediators in the inflammatory response set off the osteogenic properties of the mutant ACVR1 receptor. Members of the TGFβ superfamily, such as BMP and Activin A, which directly interact with the FOP ACVR1 receptor, mediate some of these inflammatory responses [12,22]. Our study in a human in vitro model shows that TGFβ1 is a stimulant of Activin A production in FOP. TGFβ1 significantly upregulated *INHBA* expression and Activin A production in FOP dermal fibroblasts. Given that Activin A is considered the major driver of HO in FOP [17], this finding can be of clinical significance. We also found that the FOP fibroblasts produced higher levels of Activin A compared to control cells under basal culture conditions and the initial steps towards osteogenic transdifferentiation.

Given the strong induction of Activin A by TGFβ1, we sought in turn to determine which cytokines may regulate the latter. *TGFB1* expression has been demonstrated to be increased in FOP M2 anti-inflammatory macrophages following lipopolysaccharide stimulation; in an unstimulated state, these cells also displayed significantly elevated TGFβ1 production [23]. Treatment of an FOP mouse model expressing a constitutively active mutant form of ACVR1 with an antibody against TGFβ1 was able to significantly decrease HO progression [24]. The same study identified elevated serum TGFβ after trauma induction in a HO mouse model, which coincided with the recruitment of mesenchymal stromal/progenitor cells at the injury site. Furthermore, in mice with no TGFβ1 production in neutrophils and cells of the macrophage/monocyte lineage, injury failed to trigger HO. This is in line with our findings of TGFβ being an important FOP-specific stimulant of Activin A. Interestingly, in mesenchymal stem cells derived from induced pluripotent stem cells (iMSC), treatment with TGFβ1 for 16 h in cells with the *ACVR1* mutation did not induce downstream BMP signaling pathways, suggesting that TGFβ1 might not induce Activin expression in all cell types [25]. To summarize, BMP4 appears to be a potent inducer of TGFβ1 in both FOP and control cells, while Activin A production appears more pronounced in FOP cells after TGFβ and BMP4 treatment (Figure 6).

No induction of Activin A production was observed by cytokines TNFα and IL6 despite previous findings of their upregulation in human-induced FOP macrophages [26] as well as in post-injury lesions and mast cells in the Acvr1^cR206H/+^ FOP mouse model [7]. However, the FOP fibroblasts themselves did produce more Activin A than control fibroblasts under basal culture conditions. Given the multipotent plasticity and variability of fibroblasts [27], it can be expected that dermal fibroblasts are able to simulate the nature of the mesenchymal stem progenitors in the FOP lesions. In the early stages of a flare-up, the lesion is known to be infiltrated by B and T lymphocytes and mast cells, likely delivering the Activin A to the site of inflammation [7,28,29,30,31], which in turn triggers the FAPs to commit to the osteogenic lineage. Indeed, recent single-cell analysis performed in murine FOP-like lesions has revealed how macrophages and fibroblast-like cells are the main sources of local Activin A in damaged muscles, prior to HO [11,32]. Our findings further support the notion that Activin A production may be even further propagated by the mesenchymal stem cells themselves in the HO area.

It is noteworthy that the fibroblasts that showed the highest production of Activin A were derived from patients experiencing a flare-up. Inflammation is known to induce epigenetic changes, which can influence the expression of genes such as *INHBA* [33]. FOP dermal fibroblasts were shown to react to Activin A based on phosphorylation of SMAD1/5/9 and expression of *ID1* and *ID2* in combination with higher potential for osteogenic transdifferentiation [19], and no difference between the control and FOP cells were observed in phosphorylated SMAD1/5/9 and BMP target gene expression after BMP stimulation. Similarly, in FAPs, no difference in phosphorylated SMAD1/5/9 was seen between *Acvr1^R206H/+^* and controls after BMP stimulation. However, canonical SMAD1/5/9 activation does not seem different in FOP monocytes compared to controls [3,23]. Despite this, FOP monocytes and macrophages exhibit a distinct proinflammatory secretome (including TGFβ), suggesting that alternative pathways, other than SMAD1/5/9 activation, can mediate FOP cell dysfunction.

Latent TGFβ is stored within the bone matrix and can be activated by the resorption of the calcified cartilage during endochondral remodeling [24]. TGFβ exists in three isoforms: TGFβ1, TGFβ2, and TGFβ3. TGFβs are expressed with latency-associated proteins, rendering them often inactive in the extracellular matrix in different tissues. TFGβ1, TFGβ2, and TFGβ3 are key modulators of tissue healing after injury, with one isoform showing overlapping, and also separate effects [34]. TGFβ1, the most prevalent isoform, is often involved in the promotion and differentiation of stem cells. For instance, in bone, TFGβ1 is involved in the proliferation of osteoblastic cells, further inducing bone formation [34]. In an FOP-iPSC model wherein a BMP-specific luciferase reporter construct (BRE-Luc) was transfected, no increase or difference in BMP signaling was found between the different TGFβ isoforms [25]. However, TGFβ3 did enhance chondrogenesis of 3D chondrogenic micro mass formation in both FOP and control cells [25]. Similarly, when investigating the effect of the separate TGFβ isoforms in a C2C12 myoblast line, TGFβ increased proliferation but not in an isoform-dependent manner. It will be intriguing to clarify the role of chondrogenic stimuli by the different TGFβ isoforms. For this new cell, models mirroring the endochondral ossification in FOP will need to be developed. Our model encapsulates the osteogenic transdifferentiation process, but during this process no increased expression of chondrogenic markers was detected, suggesting the endochondral aspect is not well reflected in this model [20].

Given the complex nature of inflammation, it is equally important to determine cross-reaction and synergistic effects with numerous other growth factors in the FOP HO microenvironment and their effect on the HO progenitor cell type(s). It is suggested that flare-ups are characterized by a catabolic stage with tissue destruction and infiltration of immune cells, which is followed by endochondral bone formation [35]. TGFβ can be delivered by immune cells but is thought to be activated during the resorption that takes place in cartilage remodeling. Targeting TGFβ may present a therapeutic window at different stages of HO. There is already experience inhibiting TGFβ as a potential therapy for various diseases. In some forms of cancer, TGFβ signaling contributes to cancer cell proliferation and metastasis. Multiple clinical trials have been performed to inhibit these processes with various TGFβ inhibitors, and although clinical efficacy in these trials has often been disappointing, the treatment itself was generally well tolerated [36]. In the field of bone diseases, Fresolusimab, an antibody neutralizing all TGFβ isoforms, has been tested in a phase 1 trial in Osteogenesis Imperfecta patients, exploring its effects on bone remodeling and bone density. Higher dosages resulted in sustained suppression of bone turnover, and increased bone mineral density and side effects were deemed acceptable [37]. However, long-term data are lacking, and given the pleiotropic signaling of TGFβ, lifetime inhibition of TGFβ may result in toxicity. Systemic suppression of TGFβ may have an impact on wound healing, tissue repair, and inflammatory responses [34]. In a conditional knockout model of TGF-βR2 in renal tubular cells, increased renal inflammation was noted [38]. Though, in soluble TGF-receptor IgGFc chimera transgenic mice, lifetime blocking of TGFβ had no discernable negative side effects [39]. At the time of writing, no specific TGFβ inhibitors have been FDA or EMA approved for clinical use, but given the vast experience with some TGFβ inhibitors, perhaps these could be appropriated for evaluation in a clinical FOP trial after further preclinical evaluation in FOP-specific cell and animal models.

To conclude, the findings of this study revealed TGFβ1 as an upstream FOP-specific inducer of Activin A production in patient-derived fibroblasts; Activin A and TGFβ, in turn, were both upregulated by BMP4 irrespective of mutational *ACVR1* status (Figure 6). Thus, TGFβ may present an important complementary target next to Activin A, blocking of which may (at least partially) prevent the molecular cascade culminating in Activin A-mediated FOP HO. This study was performed in FOP dermal fibroblasts, which can be expected to mirror comparable characteristics of multipotent FOP progenitor cells. We await with excitement future findings about the cell populations in which this mechanism is reflected in FOP in order to provide orientation regarding the development of additional therapeutic modalities for this debilitating disease.

## 4. Materials and Methods

### 4.1. Clinical Characteristics of FOP Patients

Patients experienced their first flare-up during the first 4 years of their life [19]. At the time of the biopsy, patients were questioned and examined to determine if they were currently experiencing a flare-up and if so, which medication was being used. Patients 2, 4, and 5 received medication due to a local flare-up which was administered from at least 1 week prior to biopsy acquisition until at least 1 week thereafter (Table 1). Patients 2 and 4 received nonsteroidal anti-inflammatory drugs (NSAIDs), and although some effect was noted, the flare-up was still active during biopsy acquisition. Patient 5 received pain relief medication, as previously described [19].

### 4.2. Cell Culture

Dermal fibroblasts were isolated and cultured from a 3 mm full-thickness skin biopsy using methods previously described [19]. Fibroblasts were cultured in Ham F10 media (31550-031, Gibco, ThermoFisher, New York, US) supplemented with 10% fetal calf serum (FCS) and 1% penicillin/streptomycin (15140122, Gibco, ThermoFisher, New York, USA). Cells were maintained at 37 °C and 5% CO_2_ in humidified conditions. The osteogenic transdifferentiation was performed as previously described by Micha et al. [19].

### 4.3. DNA Analysis of Fibroblasts

200,000 cells per sample were lysed with a sequencing lysis buffer (QE09050, Biosearch technologies/Lucigen, Hoddeson, UK). Cell lysates were heated at 65 °C for 6 min, followed by heating at 98 °C for 2 min. Extracted DNA was used for PCR amplification by standard methods with a GeneAmp PCR System 9700 (Applied Biosystems, Foster City, USA) and subsequent sequencing analysis.

### 4.4. ELISA Analysis

Dermal fibroblasts were seeded at 30,000 cells per well in 24-well culture plates. Fibroblasts were serum-starved for 24 h and subsequently treated with BMP4 (AF-120-05ET, Peprotech, London, UK), Activin A (A4941, Sigma-Aldrich, St. Louis, USA), IL-6 (7270-IL-025, R&D systems, Minneapolis, USA), TNF-α (210-TA-020, R&D systems, Minneapolis, USA), or TGF-β (cat#4232-5, Biovision, Waltham, USA). After 24 and 48 h, supernatant samples were centrifuged at 600× *g* for 5 min, collected, and frozen at −20 °C. Cell supernatants were analyzed with the Human Activin A ELISA Kit (ab119568, Abcam, Cambridge, UK) and the Human TFG-β ELISA kit (ab100647, Abcam). The protocols and standard curve construction were performed according to the manufacturer’s instructions. O.D. absorbance was read at 450 nm with a Synergy HT Microplate reader (BioTek).

### 4.5. RNA Isolation, cDNA Synthesis and qPCR

Cell samples (150,000 cells/well) were treated with BMP4, TGFβ, and Activin A, respectively, for 6 h after being serum-starved overnight. Cells were lysed with RNA lysis buffer (R1060-1-50, Zymo Research, Irvine, USA) and RNA was isolated using the Quick RNA kit (R1055, Zymo Research, Irvine, USA) following the manufacturer’s instructions. cDNA was synthesized with the SuperScript™ VILO™ cDNA synthesis kit (11754250, Invitrogen™, Waltham, USA). qPCR was performed with 1 pmol/µL forward and reverse primers (IDT) as listed in Table 2, cDNA and SYBR Green master mix (4887352001, Roche) by standard methods with the Lightcycler 480 (Roche, Basel, Switsersland). The relative expression was then calculated with the LightCycler 480 release 1.5.0 SP4 software (Roche, Basel, Switserland) using YWHAZ as a housekeeping gene.

### 4.6. Protein Isolation and Western Blotting Analysis

The fibroblasts were seeded with 150.000 cells/well and treated for 60 min with Activin A or 90 min with BMP4 after serum starvation overnight. Whole-cell lysates were prepared by lysing cells in NuPAGE^®^ LDS Sample Buffer with 10% NuPAGE^®^ reducing agent. Proteins were separated in NuPAGE 4–12% BT gels using the XCell SureLock™ system and were subsequently transferred by using the iBlot Dry Blotting system (Invitrogen). Nitrocellulose membranes were blocked in the Odyssey blocking buffer (Westburg). Immunoblotting was performed in Odyssey blocking buffer with 0.1% Triton X-100.

Primary antibodies against phospho-SMAD3 (cat#52903, Abcam), phospho-SMAD1/5/9 (AB3848-I, Sigma-Aldrich), and Actin (Cat#ab14128, Abcam) were used for overnight incubation. Secondary antibody incubation was carried out for 1 h with the IRDye 800 CW goat anti-rabbit IgG and IRDye 680 CW goat anti-mouse (LI-COR Biosciences). Fluorescence was visualized by the Odyssey system equipped with the Odyssey version 4 software (LI-COR Biosciences).

### 4.7. Statistics

Analysis of variance (2-way ANOVA) was used with the factors: genotype and treatments. An adjusted *p*-value of <0.05 was considered significant for all analyses. All statistical analyses were performed using GraphPad Prism 9 software (Insight Partners, New York, USA). Data are represented as means with error and SD, mean with SD, or summary data.

## Figures and Tables

**Figure 1 ijms-24-02299-f001:**
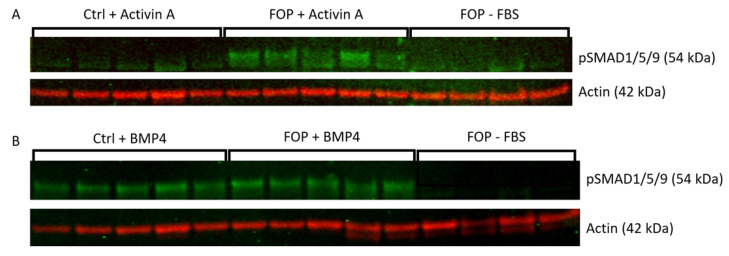
Expression of pSMAD1/5/9 in fibroblasts of FOP patients and controls by western blotting analysis. (**A**) Cells were serum-starved overnight before 60 min of stimulation with Activin A and (**B**) serum-starved overnight before 90 min of stimulation with BMP4. Actin was used to determine equal protein loading. −FBS are serum-starved unstimulated fibroblasts.

**Figure 2 ijms-24-02299-f002:**
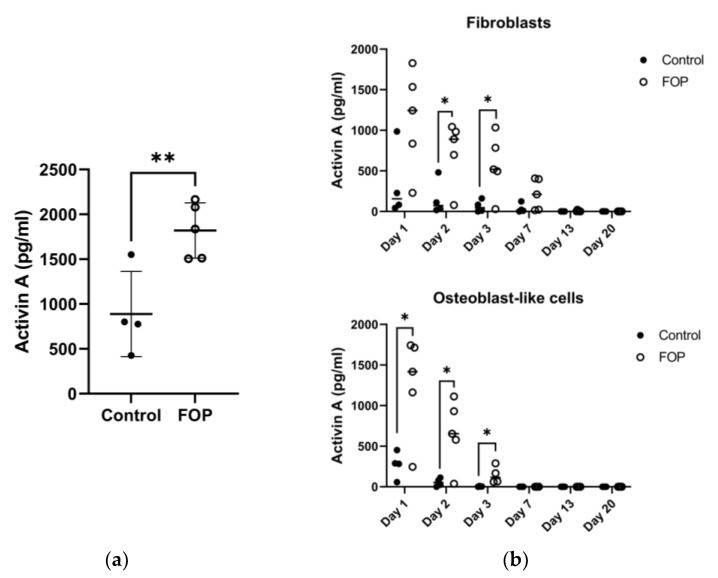
Production of Activin A as measured by ELISA in control and FOP fibroblasts. (**a**) After 24 h of seeding, Activin A was measured in the cell supernatant. (**b**) Activin A was measured on day 1, 2, 3, 7, 13, and 20 in fibroblasts and in osteoblast-like cells during osteogenic transdifferentiation. Data indicates Activin A production per cell line and their mean. * *p* < 0.05 and ** *p* < 0.005 as determined by standard student’s t-test.

**Figure 3 ijms-24-02299-f003:**
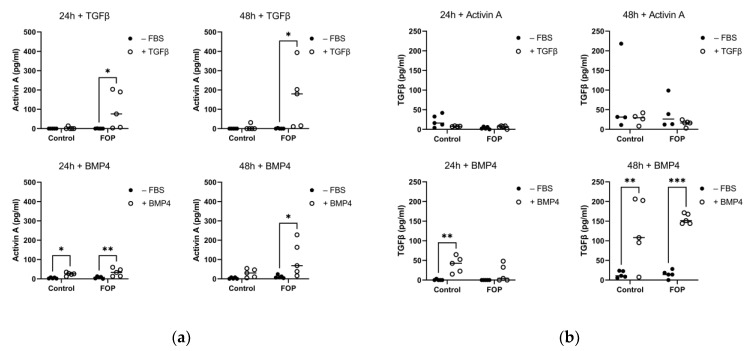
Activin A and TGFβ1 production measured by ELISA upon Activin A, TGFβ1, and BMP4 stimulation in FOP and control fibroblasts. Fibroblasts were subjected to overnight serum starvation (−FBS) before being stimulated with TGFβ1 and BMP4 for 24 and 48 h. This was followed by the measurement of (**a**) Activin A and (**b**) TGFβ1. Data is shown as the production level per cell line and their mean. * *p* < 0.05, ** *p* < 0.005, *** *p* < 0.0005 as determined by 2-way ANOVA.

**Figure 4 ijms-24-02299-f004:**
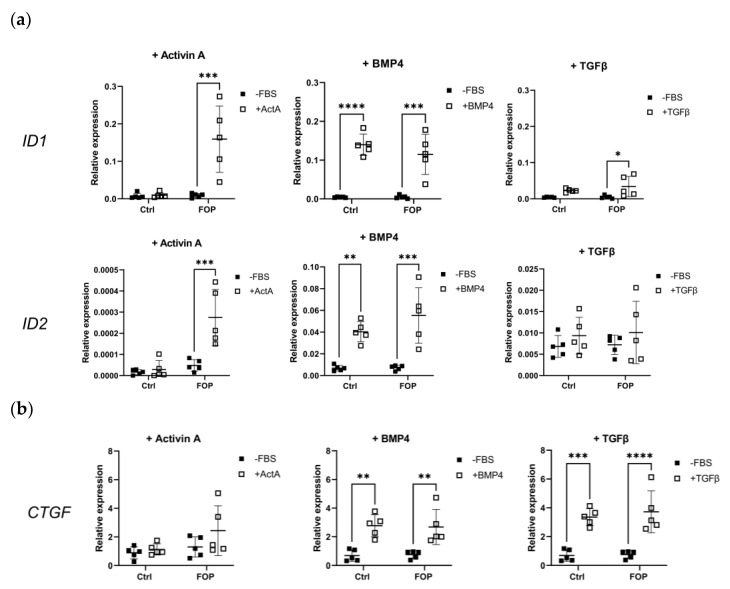
Relative gene expression of BMP and TGFβ target genes. Fibroblasts from 5 FOP and 5 control cell lines were serum-starved overnight (-FBS) before treatment with Activin A (+Activin A), BMP4 (+BMP4) and TGFβ1 (+TGFβ) for 6 h. Relative gene expression was measured by qPCR for (**a**) BMP target genes *ID1* and *ID2* and (**b**) TGFβ target gene *CTGF*; YWHAZ was used to normalize gene expression. −FBS indicates cells in medium without fetal bovine serum. Data is shown as the expression levels per cell line and their mean. * *p* < 0.05, ** *p* < 0.005, *** *p* < 0.0005, **** *p* < 0.00005 as determined by 2-way ANOVA.

**Figure 5 ijms-24-02299-f005:**
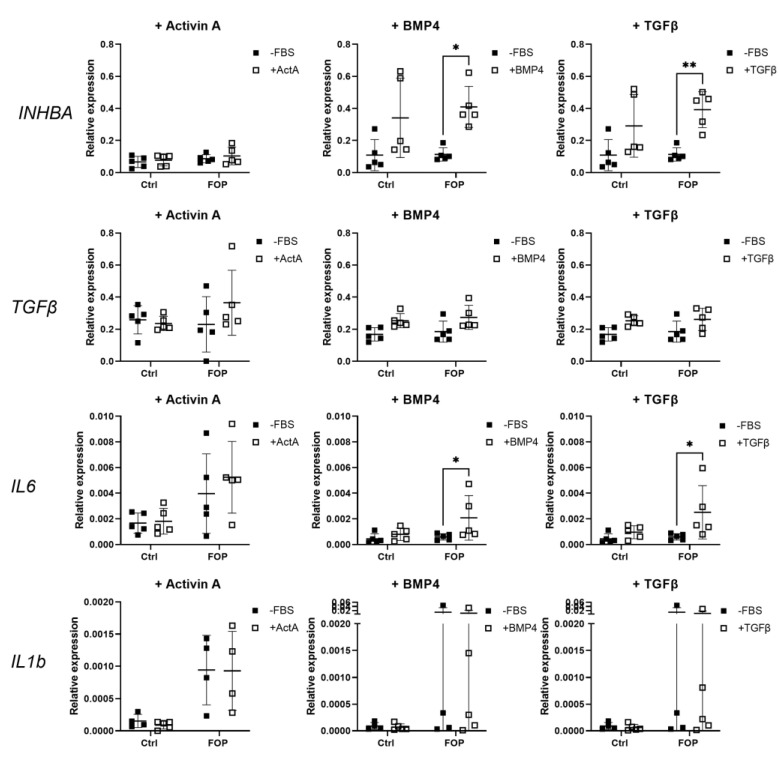
Relative gene expression of cytokines. Fibroblasts from 5 FOP and 5 control cell lines were serum-starved overnight (−FBS) before treatment with Activin A (+Activin A), BMP4 (+BMP4) and TGFβ1 (+TGFβ) for 6 h. Relative gene expression was measured by qPCR for *INHBA*, *TGFBR1*, *IL6*, and *IL1b*; *YWHAZ* was used to normalize gene expression. Data is shown as the expression levels per cell line and their mean. * *p* < 0.05, ** *p* < 0.005 as determined by 2-way ANOVA.

**Figure 6 ijms-24-02299-f006:**
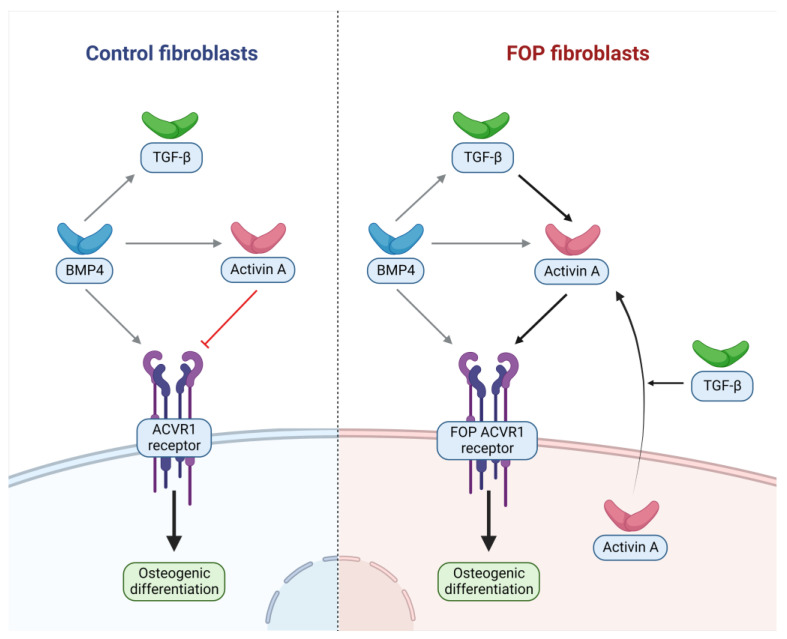
Schematic summary of main cytokine interactions with fibroblasts. TGFβ1 and BMP4 stimulation lead to increased Activin A production in FOP fibroblasts. In control fibroblasts Activin A production is increased less following exposure to BMP4 and not increased after TGFβ1 exposure. In both FOP and control, fibroblasts BMP4 leads to increased TGFβ1 production.

**Table 1 ijms-24-02299-t001:** Overview of clinical characteristics of FOP patients.

	Gender	Age at biopsy	Flare-Up < 1 Week Prior to Biopsy	Medication Used During Biopsy
Patient 1	Male	23.2	No	-
Patient 2	Female	21.9	Yes	Celecoxib
Patient 3	Male	62.4	No	-
Patient 4	Male	14.8	Yes	Naproxen
Patient 5	Female	39.6	Yes	Tramadol

**Table 2 ijms-24-02299-t002:** Primers used for qPCR analysis. Abbreviations: ID1—Inhibitor of DNA binding 1, ID2—Inhibitor of DNA binding 2, DLX1—Distal less homeobox 1, MSX2—Msh homeobox 2, CTGF—connective tissue growth factor, COL1A1—collagen type 1 alpha 1 chain, THBS1—Thrombospondin 1, INHBA—Inhibin A, TGF-β1—transforming growth factor-β1, IL-6—interleukin-6, IL-1b—interleukin-1b, YWHAZ—YWHAZ housekeeping gene.

Gene	NM Number	Sequence
*ID1*	NM_002165	AATCCGAAGTTGGAACCCCC
		AACGCATGCCGCCTCG
*ID2*	NM_002166	GTGGCTGAATAAGCGGTGTTC
		CTGGTATTCACGCTCCACCT
*DLX1*	NM_178120	GACTCACACAGACTCAGGTCAA
		AGCGGGTTTATCTTGCTGCT
*MSX2*	NM_002449	TCATGGCTTCTCCGTCCAAA
		AGGGCTCATATGTCTTGGCG
*CTGF*	NM_001901	ATTCTGTCACTTCGGCTCCC
		TCCAGTCGGTAAGCCGC
*COL1A1*	NM_000088	GTGCTAAAGGTGCCAATGGT
		ACCAGGTTCACCGCTGTTAC
*THBS1*	NM_003246	CTCAGGAACAAAGGCTGCTC
		TGGACAGCTCATCACAGGAG
*INHBA*	NM_002192	GTTTGCCGAGTCAGGAACAG
		TCACAGGCAATCCGAACGTC
*TGFβR1*	NM_000660	CCGACTACTACGCCAAGGAG
		GGTATCGCCAGGAATTGTTG
*IL6*	NM_001371096	AGTTCCTGCAGAAAAAGGCAAAG
		AAGCTGCGCAGAATGAGATGA
*IL1b*	NM_000576	AGCCATGGCAGAAGTACCTG
		CCTGGAAGGAGCACTTCATCT
*YWHAZ*	NM_001135701.2	GATGAAGCCATTGCTGAACTTG
		CTATTTGTGGGACAGCATGGA

## Data Availability

The data presented in this study are available on request from the corresponding author.

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
