# Peer review of "TGF-Beta Induces Activin A Production in Dermal Fibroblasts Derived from Patients with Fibrodysplasia Ossificans Progressiva"

_ijms, 2023, doi:10.3390/ijms24032299_

Round 1

Reviewer 1 Report

This manuscript explored TGFβ1 to be a FOP-specific stimulant of Activin A, shown by the upregulation of INHBA gene and protein expression in dermal fibroblasts derived from patients with Fibrodysplasia Ossificans Progressiva. The results showed the first time TGFβ as a triggering factor of Activin A production in FOP. Major revisions are necessary in the text throughout the paper. In addition, improve the content of the manuscript according to the following comments.

1.     The argument about the interactions with cytokine and fibroblasts was not complete, and relevant experiments such as diluciferase reporter experiment are supplemented.

2.     For this newly discovered target TGFβ, blocking experiments should be carried out to verify the conclusions.

3.     Relative gene expression of BMP and TGFβ target genes in Fig 4 and relative gene expression of cytokines in Fig 5 are only qPCR results, please supplement the corresponding ELISA results. And the expression of IL1b were not credible in FOP group because the error bar was too large.

4.     Please explain the “osteoblast-like cells” mentioned in Fig 2(b).

5.     Relative to other patients, the patient 3 is too old. Will there be an impact on the results? And whether the medication used during biopsy would affect the results?

Reviewer 2 Report

In this study the authors investigate the molecular mechanism underlying FOP related ossification using patient derived dermal fibroblast cell lines. Specifically, they pinpoint TGFB1 as a prominent stimulant of Activin A which in turn drives ossification. Based on in vitro assays, they demonstrate TGFB1 as a putative target for FOP treatment. While this study is of significant translational relevance, I have the following comments to strengthen the publication of this manuscript:

Major comments

1.     The authors comment (Lines 253-254) that “FOP is known to progress by endochondral ossification; thus, equally intriguing is the role of chondrogenic stimuli by different TGFb isoforms”. It is pertinent to this study to show the role of TGFb isoforms 2&3 to show that TGFB1 is highly specific to FOP disease context.

2.     TGFB is a pleiotropic signaling mechanism that acts in various cell types, orchestrating multiple biological processes. Can the authors comment on potential side effect of TGFB1 as a therapeutic target?

3.     The authors comment on the role of immune cells particularly macrophages and monocytes in creating an inflammatory niche that drives ossification in FOP lesions. Can you demonstrate the impact of macrophage populations in directly driving osteogenesis in FOP dermal fibroblasts using co-culture experiments?

4.     To further strengthen the findings of this manuscript I recommend investigating changes in the transcriptome of FOP patients (with and without flare ups) pre and post TGFb treatment to assess both differential cytokine activity.

Minor comments:

1.     Please redo Figure 1A as the green signal for pSMADs is observed in the actin lane (FOP – FBS)

2.     Please perform spell checks and examine the manuscript for occasional typing errors.

Round 2

Reviewer 1 Report

All my questions answered.